# The Synergy of Chicken Anemia Virus and Gyrovirus Homsa 1 in Chickens

**DOI:** 10.3390/v15020515

**Published:** 2023-02-13

**Authors:** Mengzan Yang, Qi Yang, Xiaoqing Bi, Hengyang Shi, Jianhao Yang, Xiangyu Cheng, Tianxing Yan, Honghai Zhang, Ziqiang Cheng

**Affiliations:** College of Veterinary Medicine, Shandong Agricultural University, Tai’an 271018, China

**Keywords:** chicken anemia virus (CAV), Gyrovirus homsa 1 (GyH1), synergy, co-infection, immunosuppression, pathogenicity, viral replication

## Abstract

Chicken anemia virus (CAV) and Gyrovirus homsa 1 (GyH1) are members of the Gyrovirus genus. The two viruses cause similar clinical manifestations in chickens, aplastic anemia and immunosuppression. Our previous investigation displays that CAV and GyH1 often co-infect chickens. However, whether they have synergistic pathogenicity in chickens remains elusive. Here, we established a co-infection model of CAV and GyH1 in specific pathogen-free (SPF) chickens to explore the synergy between CAV and GyH1. We discovered that CAV and GyH1 significantly inhibited weight gain, increased mortality, and hindered erythropoiesis in co-infected chickens. Co-infected chickens exhibited severe immune organ atrophy and lymphocyte exhaustion. The proventriculus and gizzard had severe hemorrhagic necrosis and inflammation. We also discovered that the viral loads and shedding levels were higher and lasted longer in CAV and GyH1 co-infected chickens than in mono-infected chickens. Our results demonstrate that CAV and GyH1 synergistically promote immunosuppression, pathogenicity, and viral replication in co-infected chicken, highlighting the interaction between CAV and GyH1 in the disease process and increasing potential health risk in the poultry breeding industry, and needs further attention.

## 1. Introduction

Gyroviruses (GyVs) of the *Anelloviridae* family are small, non-enveloped, icosahedral with a negative-sense, circular, single-stranded DNA (ssDNA) genome of approximately 2.3–2.5 kb. Originally, the Gyrovirus was classified as a member of the *Circoviridae* family due to its ssDNA structure [1], but later reassigned to the *Anelloviridae* family in 2017 by the International Committee on Taxonomy of Virus (ICTV) with the differences in the genomic characteristics of *Circoviridae* members [2]. Initially, the *Gyrovirus* had only one officially recognized species, the chicken anemia virus (CAV). Numerous diverse viruses related to CAV have been identified but not classified during the past decade [3,4,5,6,7,8,9,10]. Nine new species have been established to accommodate the unclassified gyroviruses until 2021 [11]. There are three partially overlapping open reading frames in the genome *Gyroviruses*, which encode capsid structural protein VP1, scaffold protein VP2, and apoptosis protein VP3. Among them, VP1 has a unique role to play in the virus independence, assembly, replication and infection. Thus, VP1 is often used in order to detect antigens.

CAV has been widely distributed and prevalent worldwide since its discovery in 1979 [12,13]. Chicks aged 2–4 weeks and those lacking maternal antibodies are the main targets of CAV infection. The offspring of vaccinated breeders produce maternal antibodies to protect them from CAV. However, adult chickens can also be affected by horizontally transmission of the virus and show subclinical symptoms such as stunted growth when the maternal antibodies are weakened or affected by the feeding environment. The key target cells of CAV are hematopoietic cells of bone marrow, lymphocyte and macrophage precursor cells of the thymus, and the bursa of Fabricius. CAV invades hematopoietic cells in the bone marrow, drastically reducing the number of erythrocytes produced, resulting in host aplastic anemia. After vertical transmission from the ovary, CAV has the ability to cause immunosuppression by itself or in conjunction with other infectious agents, increasing the risk of secondary infections in chicks by exposing them to bacteria or viruses. The main clinical symptoms of chicken infectious anemia (CIA) caused by CAV are aplastic anemia, wasting, and severe immunosuppression [14]. The CIA outbreak remains a major challenge for the global poultry industry.

Gyrovirus homsa 1 (GyH1, also known as Gyrovirus 3, GyV3) was first identified in the feces of children with acute gastroenteritis in Chile [6]. Afterward, GyH1 was detected in the feces of a diarrhea patient in Hong Kong and South Africa [3,15] and the San Francisco market chicken meat [10]. GyH1 was first isolated from transmissible viral proventriculitis (TVP) broilers in China in 2018. Through genetic research, it was shown that cat-sourced GyH1 strains were identified from pet cat feces in northeast China in 2019 and shared a high degree of nucleotide and amino acid similarity with chicken-sourced GyH1. This suggests that the strain may have originated from chickens infected with GyH1 [16]. Recently, studies have revealed that the target cells of GyH1 include myeloid cells, lymphocytes, the muscle fiber cells of gizzard, etc. GyH1 infection in chickens causes aplastic anemia, immunosuppression, and TVP [17]. Sero-epidemiological investigation indicates that GyH1 infection is widespread in chicken farms in China [18].

Infection with more than one microbe (protozoa, virus, bacteria, etc.) is called a mixed infection. In virology, coinfection is used to describe simultaneous infection of a cell or organism by separate viruses. Co-infection contributes largely to affecting disease severity and epidemiology and is commonly found in nature. The outcome of any co-infection is complex and various factors influence it, such as the interactions between the pathogens, the characteristics of the target cells, and the induced immune response [19]. The effect of one virus on the replication of another is known as viral interference, and it is the most common outcome of co-infection, in which one virus competitively constrains the replication of the other, or promotes other viruses’ replication within the host cell.

Co-infection of viruses is ubiquitous in poultry. According to accumulated evidence, co-infection with several immunosuppressive viruses is more likely to cause secondary infections in chickens, resulting in severe pathogenic effects. Co-infected chicks with CAV and avian reovirus have exhibited severe reductions in packed cell volume and tissue damage [20]. Additionally, the synergistic pathogenic effect of avian leukosis virus subgroup J (ALV-J) and avian reticuloendotheliosis virus (REV) co-infection in chickens leads to higher mortality and susceptibility to *E. coli* infection [21]. Additional studies have revealed that ALV-J and infectious bursal disease virus (IBDV) synergistic viral replication induction can lead to severe immunosuppression in chickens and thus enhance pathogenicity [22].

Recent studies have indicated that CAV and GyH1 co-infection frequently occur in poultry production [23,24]. However, the CAV and GyH1 synergy has been uninvestigated. The current study aims to establish a co-infection animal model of CAV and GyH1 to explore the synergistic pathogenicity in chickens.

## 2. Material and Method

### 2.1. Virus and Animals

This study was performed following the regulations stipulated by the guide for the care and use of laboratory animals of the National Research Center for Veterinary Medicine. GyH1 strain SDAU-1 (log_10_ EID_50_ = 2.5; GenBank accession no. MG366592) and CAV strain Cux-1 (log_10_ EID_50_ = 3; GenBank accession no. MN079036) were serially passaged in specific pathogen-free (SPF) chickens and stored in our laboratory. SPF embryos were purchased from the Shandong Academy of Agricultural Sciences (Jinan, China). After hatching, the chicks were raised separately in four negative-pressure isolators. The temperature, humidity, and other feeding environment were kept consistent.

### 2.2. Experimental Infection and Sampling

A total of 140 SPF chicken embryos were randomly divided into four groups (35 chickens per group). The virus was inoculated by allantoic cavity inoculation at day 6 of embryonic (Ed) development. Mock chickens were inoculated with 0.2 mL phosphate buffer saline (PBS). CAV or GyH1 mono-infected embryo were inoculated with 100 EID_50_ of the CAV or GyH1 in 0.2 mL PBS. Co-infected embryos were inoculated with 100 EID_50_ of CAV and GyH1 in 0.1 mL PBS. Chickens were clinically inspected, and weights and deaths were recorded daily. CAV or GyH1 positive infection rate in the sera was detected using enzyme-linked immunosorbent assay (ELISA) seven days post-infection (dpi). Uninfected chickens were culled. Venous blood samples were collected in vacuum tubes at seven-day intervals, and then isolated sera were utilized for antigen tests. Anticoagulant-treated blood was used to prepare blood smears for hemograms and routine blood examinations. Three chickens per infected group were euthanized and necropsied at 7, 14, 21, 28, and 35 dpi. After sacrifice, a complete necropsy was performed, and gross lesions were observed. Consequently, bone marrow, thymus, bursa of Fabricius, spleen, proventriculus, and gizzard were sampled for pathogenicity and viral load testing. The necropsy tissue samples were fixed in 10% formaldehyde and embedded in paraffin wax for standard processing. After hematoxylin and eosin (HE) staining, 4 µm tissue sections were examined.

### 2.3. ELISA

The CAV-VP1 and GyH1-VP1 antigens were detected using ELISA. CAV was detected using the ELISA kit of China Homiao Biotechnology Co., LTD. Moreover, in accordance with the manufacturer’s instructions for testing. GyH1 was detected using a double-antibody sandwich ELISA (DAS-ELISA) system for antigen detection. A 96-well ELISA plate was diluted with purified GyH1-VP1 monoclonal antibody in coated buffer solution (CBS, pH 9.4) to a final concentration of 3 µg/mL, overnight at 4 °C; skim milk closed for 1 h at 37 °C; washed with phosphate buffered saline with Tween (PBST); GyH1-positive serum diluted to 1:4 with PBS was incubated for 1 h. Then, the plates were treated with 100 L of detection antibody diluted to 6 g/mL for one hour at 37 °C; negative serum and positive serum were added to each well as control; washed with PBST; 3,3′,5,5′-tetramethylhydrazine (TMB) mixed with the board for color development at 37 °C for 15 min; used 2 mol/L H_2_SO_4_ to stop color development. Optical density (OD) values were measured at 450 nm.

### 2.4. Quantitative Real-Time PCR (qPCR)

According to the manufacturer’s instructions, viral DNA was extracted from tissues using a commercial kit (TIANGEN, Beijing, China). The viral load was detected using a dual quantitative TaqMan real-time PCR system for CAV and GyH1 constructed in our laboratory. The reactions contained 10 μL Premix Ex Taq (Probe qPCR, 2×) (TaKaRa Bio, Dalian, China), 0.4 μL forward primer (10 μmol/L), 0.4 μL reverse primer (10 μmol/L), 0.8 μL probes (10 μmol/L), 1 μL template DNA, and sterile water to bring the final volume to 20 μL. The amplification parameters were 95 °C for 30 s, 40 cycles of 95 °C for 20 s, 62 °C for 30 s, and 50 °C for 30 s. Primers and probes are presented in Table 1.

### 2.5. Statistical Analysis

All data were reported as means ± standard deviation. The SPSS statistical software package for Windows version 17.0 (SPSS, Inc., Chicago, IL, USA) was used to determine statistically significant differences with two-tailed unpaired Student’s *t*-tests. *p*-values less than 0.05 were considered statistically significant.

## 3. Results

### 3.1. Clinical Manifestations

Six-day-old SPF chicken embryos were infected with CAV, GyH1, and both viruses (CAV + GyH1) to understand the co-pathogenicity of CAV and GyH1, while the Mock chickens were inoculated with PBS. Figure 1A illustrates the experimental process. The infected chickens were emaciated with messy feathers. The co-infected chickens exhibited serious growth retardation, lethargy, and depression (Figure 1B). The average body weight of CAV, GyH1 mono-infected, and CAV + GyH1 co-infected chickens at each time point was lower than that of uninfected chickens. Co-infected chickens lost more body weight than mono-infected chickens from 14 to 35 dpi. The average body weight loss was extremely significant at 21 dpi: the average weight of CAV-, GyH1-, and CAV + GyH1-infected chicken was 126 g, 142 g, and 105 g, respectively. Co-infected chicken weight was 16.6% and 26.1% lower than that of CAV and GyH1 mono-infected chickens, respectively (Figure 1C). The co-infected chicken mortality rate was higher than that of mono-infected chickens. The CAV and GyH1mono-infected chickens had an overall mortality of 15% and 25%, respectively, while the co-infected chickens had 67.5% (Figure 1D). These data demonstrate that CAV and GyH1 co-infection causes severe growth retardation and increased mortality.

### 3.2. Viremia and Viral Shedding

Viremia and viral shedding were detected in the infected chickens to investigate the influence of co-infection on infection and transmission capacity of the two viruses. The results show that the positive rate of CAV-VP1 and GyH1-VP1 in serum of co-infected chicken viremia did not decrease significantly in the chicken population at 21 and 28 dpi, while CAV-VP1 or GyH1-VP1 serum positive rates decreased in CAV or GyH1 mono-infected chickens (Figure 2A,B). CAV or GyH1 viral shedding in mono-infected chickens started at 14 dpi, peaked at 21 dpi, and began to reduce at 28 dpi, while the CAV viral shedding in co-infected chickens was significantly higher than that in CAV mono-infected chickens during the experimental period, especially at 14 dpi. The virus shedding level of GyH1 from co-infected chickens was consistently higher than that of GyH1 mono-infected chickens from 21 dpi onward (Figure 2C). CAV shedding levels peaked at 14 dpi in co-infected chickens and remained higher than in the CAV mono-infected group until 35 dpi (Figure 2D). The data demonstrate that CAV and GyH1 co-infection can cause severe and persistent viremia and virus shedding.

### 3.3. Erythrocyte Examination in the Blood

To further understand whether CAV and GyH1 co-infection can synergistically induce more severe erythrocyte disease, blood smears were utilized for observing red blood cell morphology, and blood routine (also known as complete blood count, CBC) was used in order to detect the number of erythrocytes. There were no proerythrocytes in the peripheral blood of normal chickens. GyH1 mono-infected chickens had a proerythrocyte percentage of 0.9%, CAV mono-infected birds had a proerythrocyte percentage of 1.2%, and co-infected chickens had a proerythrocyte percentage of 7.5% in their blood. The proerythrocyte proportion increased in the infected chickens, while co-infected chickens were more serious (Figure 3A). According to the dynamic changes of different mean erythrocyte counts, the number of erythrocytes of mono-infected chickens decreased at 14 dpi, increased at 21 dpi, and then tended to normal at 35 dpi. The mean number of erythrocytes did not recover in co-infected groups after 14 dpi. The erythrocyte count was 1.22 × 10^12^/L at 21 dpi, making the chicken severely anemic. The anemia did not alleviate until 21 days of age and was lower than the other groups at all time points. (Figure 3B). These data demonstrate that CAV and GyH1 co-infection cause serious anemia.

### 3.4. Dynamic Changes in the Immune Organ Index

The spleen (Figure 4A), thymus (Figure 4B), and bursa of Fabricius immune indexes (Figure 4C) were significantly lower in co-infected chickens than in CAV or GyH1 mono-infected chickens, indicating that co-infection can significantly inhibit the development and growth of immune organs. Additionally, most immune organ indexes in CAV or GyH1 mono-infected chickens began to increase or stopped decreasing, while the immune indexes of the co-infected group recovered slowly after 21 dpi. Therefore, the immune organ indexes were significantly lower in co-infected chickens than in CAV or GyH1 mono-infected chickens until 35 dpi. The above data demonstrate that co-infection of CAV and GyH1 caused serious atrophy of immune organs.

### 3.5. Viral Loads in Various Tissues at Different Time Points

The virus copy number in the proventriculus, gizzard, spleen, bursa of Fabricius, bone marrow, and thymus was detected using qPCR to analyze the CAV and GyH1 viral loads in infected chicken tissues. The GyH1 viral load in GyH1 mono-infected and co-infected chickens did not differ significantly in any tissue at 7 dpi. The GyH1 viral load of co-infected chickens was higher than that of mono-infected chickens at 21 dpi. The gizzard, bone marrow, and thymus of co-infected chickens had significantly higher GyH1 viral loads than GyH1-infected chickens at 21 dpi and 35 dpi, but no significant difference in other organs (Figure 5A–C). The CAV copy number in co-infected tissues was higher than in CAV-infected chickens at 7 dpi, and this difference was extremely significant at 21 dpi and 35 dpi (Figure 5D–F). These results show that GyH1 promotes CAV replication in chickens. The data demonstrate that CAV can significantly promote the GyH1 replication in the gizzard, bone marrow, and thymus at 21 dpi, and GyH1 can significantly enhance the CAV replication in all tissues detected at each stage of infection.

### 3.6. Histopathological Results

Tissues from different infected chickens were embedded in paraffin for histopathological observation to understand better the functional damage of immune and digestive organs caused by the co-infection with CAV and GyH1. The results presented that co-infected chickens have the most severe lymphocyte loss in their immune organs. The hematopoietic and myeloid cells were replaced by adipose tissue in bone marrow in CAV and GyH1 mono-infected chickens. The bone marrow stromal cell structure was severely damaged in co-infected chickens, and the island of young erythrocytes was completely lost due to the more severe hyperplasia of adipocytes in bone marrow. The digestive organ damage was more severe in co-infected chickens, manifested in the necrosis of proventriculus tubules and the bleeding necrosis of the gizzard (Figure 6). The data demonstrate that the CAV and GyH1 co-infection intensifies immunosuppression and tissue damage.

## 4. Discussions

Although CAV eradication has been continuously carried out, the prospects for CAV prevention and control are still pessimistic due to its strong transmission capacity [25]. Additionally, no reliable commercial vaccine has been developed, even though the serious harm of GyH1 to poultry farms has been gradually recognized with in-depth research on GyH1. Recent clinical cases of CAV and GyH1 co-infection have been discovered in chicken farms in Shandong Province, China [23]. Mortality and disease severity was greater in CAV and GyH1 co-infected chickens than in CAV or GyH1 mono-infected chicken. However, the synergistic pathogenicity of CAV and GyH1 in chicken remains obscured. CAV and GyH1 co-infected chickens exhibited more severe growth inhibition, atrophy of immune organs, and increased mortality.

CAV and GyH1 are primarily transmitted vertically using eggs, but could be transmitted horizontally through bedding, feed, air, and other media too. These two viruses are highly contagious. The virus persistence in the environment is closely related to the virus-shedding dynamics [26]. Our experiment proves that CAV and GyH1 co-infected chickens exhibited increased and prolonged virus shedding, which may enhance its transmission capability between chickens. According to Gonzalez’s extended propagation experiment, the H7N1 highly pathogenic avian influenza virus (HPAIV), causing a large-scale epidemic in commercial chicken farms in Italy, has a higher level of cloacal shedding than the low pathogenic precursor virus (LPAIV), and thus has a higher chance of contaminating the chicken farm environment (such as worker soles, feed), making indirect and direct transmission of the virus between flocks more effective [27]. A higher level of CAV or GyH1 shedding in co-infected chickens is likely to increase the virus transmission in chickens, making disease control difficult.

Most viruses associated with aplastic anemia (AA) are non-cytopathic or poorly cytopathic for hematopoietic stem and progenitor cells (HSPCs). They tend to persist in infecting their hosts [28]. The number of erythrocytes in CAV and GyH1 co-infected chickens decreased sharply, which can be superficially explained by the fat replacement of erythroblasts islands in the bone marrow. However, the potential synergistic mechanism that seriously impacts HSPCs still needs further study. Recent evidence proved that cytomegalovirus (CMV) and Epstein–Barr virus (EBV) co-infection leads to more severe AA, which is associated with T cell-mediated inhibition of erythroid colony-forming unit proliferation in bone marrow [29]. CAV and GyH1 co-infection may also induce erythroid progenitor cell differentiation blocking and apoptosis through this pathway, resulting in a sharp decline or cessation of erythropoiesis.

Experiments have confirmed that when one or both immunosuppressive viruses infect the same host, the severity of immunodeficiency increases. ALV-J and REV/IBDV/MDV co-infection of chickens can lead to severe immune organ dysplasia and T and B lymphocyte exhaustion [21,30,31]. In our experiment, the decrease in the immune organ index of the bursa, thymus, and spleen in CAV and GyH1 co-infected chickens intuitively reflected the degree of immune tissue atrophy. Furthermore, histopathological examination of the immune organs of CAV and GyH1 co-infected chickens revealed severe lymphocyte depletion and hyperplasia of fibrous connective tissue. These severe lesions are likely to cause more severe immunosuppression. Studies have confirmed that porcine pseudorabies virus (PRV) and porcine circovirus type 2 (PCV2) co-infection inhibits signal transduction in cytokine expression pathways, such as IFN-β and TNFα, thereby inhibiting the immune response [32]. CAV and GyH1 co-infection may exacerbate immunosuppression by blocking the cytokine expression pathway, but the specific mechanism needs further exploration.

In this study, the CAV load was significantly higher in co-infected chickens than in CAV-infected chickens in each organ at all the detection time points, indicating that GyH1 can significantly promote CAV replication in chickens. Previous studies have demonstrated that porcine reproductive and respiratory syndrome virus (PRRSV) and PCV2 co-infection have enhanced viral replication by inducing pro-inflammatory responses and reducing the defense function of alveolar macrophages in pigs [33]. Additionally, co-infected chickens had significantly higher GyH1 loads than GyH1-infected chickens in the gizzard, bone marrow, and thymus, suggesting that the synergistic viral replication occurs in an appropriate environment. As shown in the GyH1 viral loads, CAV showed poor enhancement of GyH1 replication. We speculate that it may be because there is a competitive relationship between CAV and GyH1 replication in the early stage. This interesting phenomenon requires further study. We also discovered that the greater the viral load, the more severe the tissue damage in co-infected chickens, manifested as shrinkage of the spleen, thymus, and bursa, extensive bleeding, and erosion of the proventriculus and gizzard (Appendix A). According to previous research, there is an inextricable link between the deterioration of organ lesions and an increase in viral load in human immunodeficiency virus (HIV) and torque teno virus (TTV) co-infected patient tissues [34].

In summary, this is the first study to demonstrate a synergy between CAV and GyH1 in co-infected chickens, as reflected by the fact that CAV and GyH1 co-infection induced a more severe impact on the development of immune organs and pathogenicity of multiple organs of chickens. This study opens a new field for the research of synergistic pathogenesis. It provides a warning for CAV and GyH1 prevention and control and the necessity of CAV or GyH1 eradication in large-scale chicken farms.

## Figures and Tables

**Figure 1 viruses-15-00515-f001:**
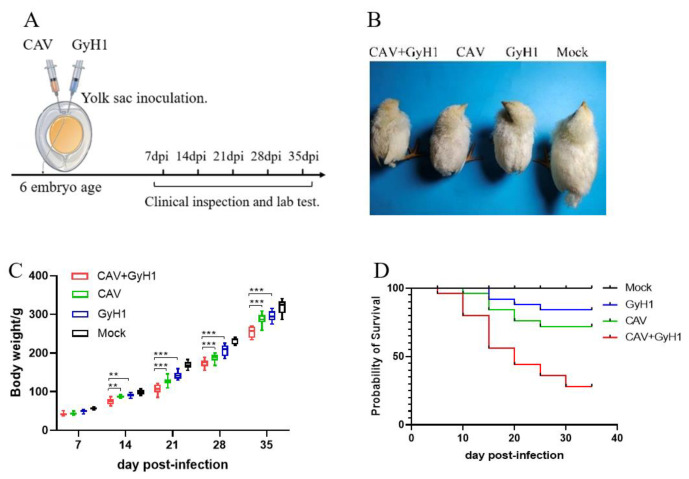
Experimental design and clinical presentation of infected chickens. (**A**) Schematic diagram of experimental infection and testing. Relevant data detection is performed at specified time points. (**B**) Clinical observation of chicken in each group. (**C**) Dynamic changes of weight gain of chickens in four groups. (**D**) Survival curve. **, *p* < 0.01; ***, *p* < 0.001, (Student’s *t*-tests).

**Figure 2 viruses-15-00515-f002:**
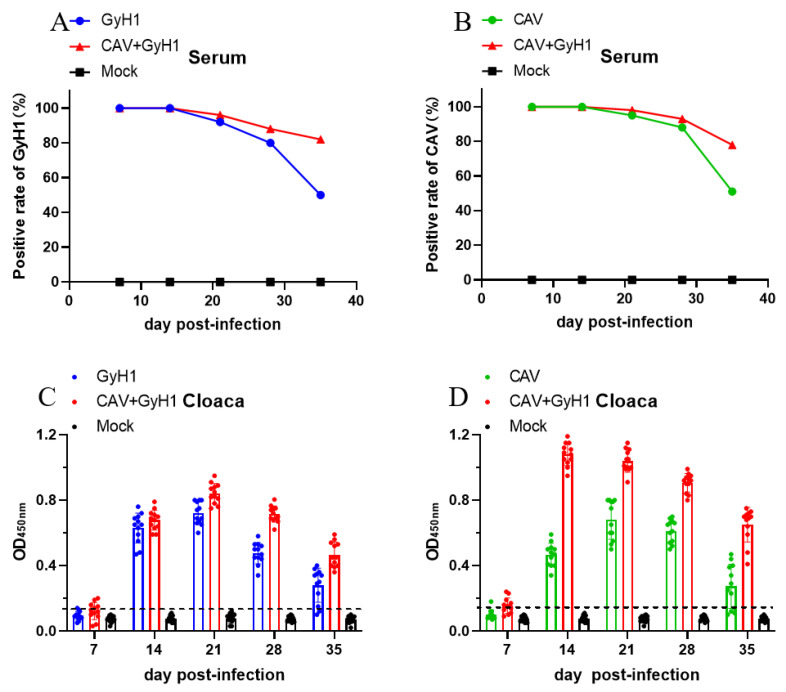
Dynamic changes of positive viremia rate and virus shedding. (**A**) Comparison of the GyH1 viral positive rates in serum in Mock, GyH1, and co-infected chickens at different time points, determined by DAS-ELISA. (**B**) Comparison of the CAV viral positive rates in serum from chickens in Mock, CAV, and co-infected chickens at different time points, determined by ELISA. (**C**) Comparison of VP1 antigen levels of GyH1 in cloacal swabs in Mock, GyH1, and co-infected chickens at different time-points, determined by DAS-ELISA. (**D**) Comparison of VP1 antigen levels of CAV in cloacal swab Mock, GyH1, and co-infected chickens at different time-points, determined by ELISA.

**Figure 3 viruses-15-00515-f003:**
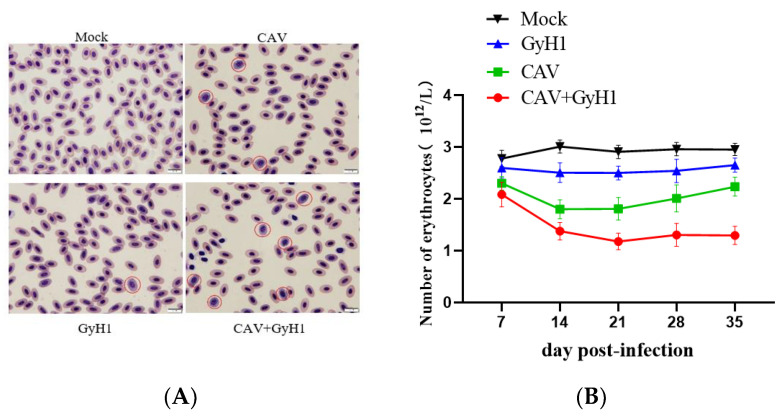
Erythrocyte morphology and count. (**A**) Erythrocytes were observed with Swiss–Giemsa staining, 100×. The rounded, basophilic and thus appearing blue is proerythrocyte, which is indicated by a red circle. (**B**) Dynamic changes of erythrocyte number in each group determined by routine analysis of blood.

**Figure 4 viruses-15-00515-f004:**
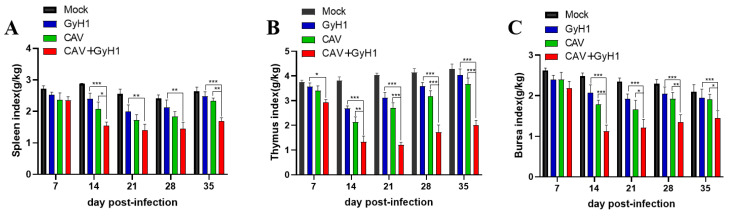
Dynamic changes of immune organ index. (**A**) Spleen index. (**B**) Thymus index. (**C**) Bursa of Fabricius index. Immune organ index = Immune organ weight (g)/live total chicken weight (kg). * , *p* < 0.05; **, *p* < 0.01; ***, *p* < 0.001, (Student’s *t*-tests).

**Figure 5 viruses-15-00515-f005:**
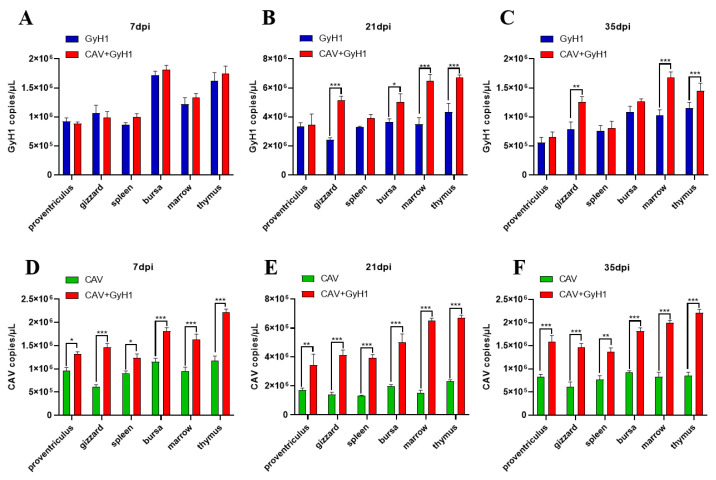
Comparison and dynamic changes of viral loads in infected chicken tissues. (**A**–**C**) Comparison of organ GyH1 viral loads between the GyH1- and co-infected chickens, determined by qPCR. (**D**–**F**) Comparison of organ CAV viral loads in CAV- and co-infected chickens, determined by qPCR. * , *p* < 0.05; ** , *p* < 0.01; *** , *p* < 0.001, (Student’s *t*-tests).

**Figure 6 viruses-15-00515-f006:**
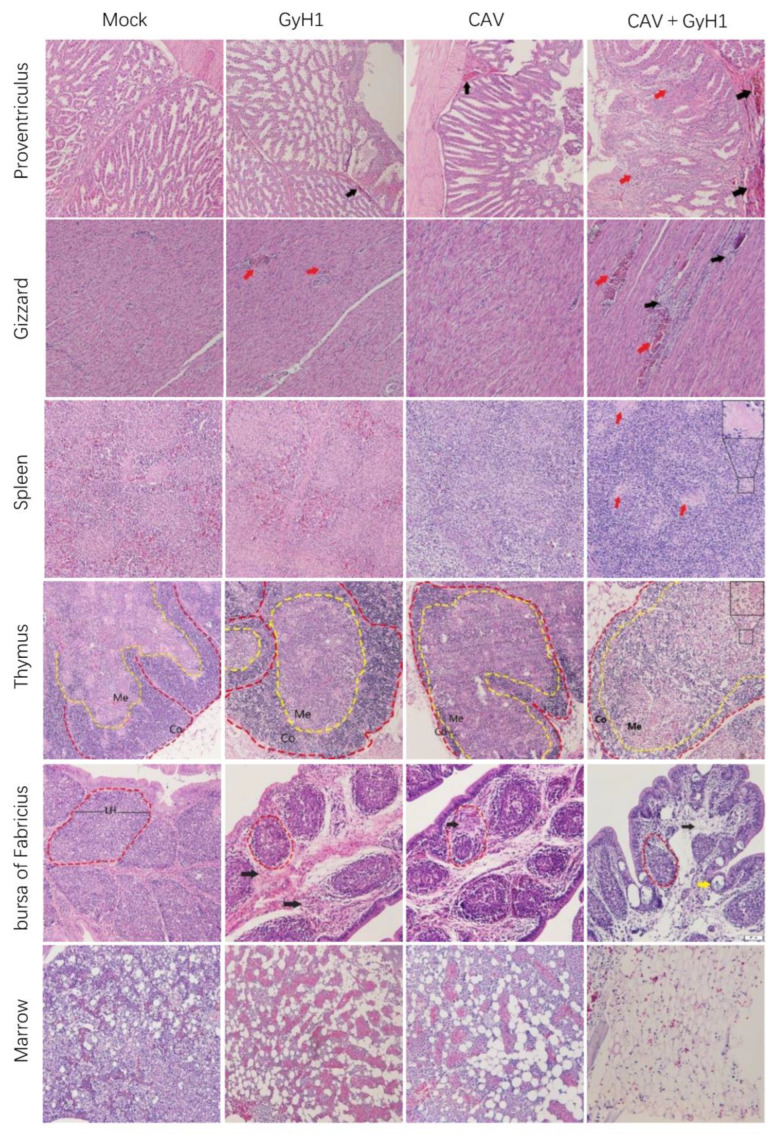
Histopathological examination of immune and digestive organs (HE staining, 20×). Mucosal lamina propria exudation of erythrocytes in the proventriculus (black arrows), and structural disturbances resulting from necrosis of glandular tubule cells (red arrows); muscular hemorrhage (red arrows) and myofiber necrosis (black arrows) in the gizzard; splenic lymphocytes were lost, and germinal centers were enlarged (red arrows); decreased thymic cortical area (Co) and enlarged medullary substance region (Me); bursal lymphoid follicles (LH) are miniaturized with connective tissue hyperplasia (black arrow) and lymphatic follicular emptying (yellow arrow); adipocytes replace myeloid cells in the bone marrow.

**Table 1 viruses-15-00515-t001:** Real-time PCR primers and probes.

Primers and Probes		Sequence (5′-3′)	Size (bp)
CAV	Forward	GGTGTTCAGGCCACCAACAA	138
	Reverse	GCAGATCTTAGCGTGGGAGC	
	Probe	FAM-TATCGCTGTGTGGCTGCGCGAATGC-BHQ2	
GyH1	Forward	GTCGGTTGGGACTCTCTCCA	139
	Reverse	GGAACCAGTGGCGTCTGAAG	
	Probe	CY5-TTCCTGGCTTCGCGAGTGTTCGCGT-BHQ2	

## Data Availability

All data relevant to this study can be found in this article.No new data were created or analyzed in this study. Data sharing is not applicable to this article.

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
