# Peer review of "The Synergy of Chicken Anemia Virus and Gyrovirus Homsa 1 in Chickens"

_viruses, 2023, doi:10.3390/v15020515_

Round 1

Reviewer 1 Report

This study is the first to examine a synergetic effect between CAV and GyH1 in chickens. Since both viruses are ubiquitous in chicken flocks, it seems important to define synergetic effects to prevent and control the diseases caused by CAV and/or GyH1. However, several parts in MS were difficult to understand. Also, there are many careless mistakes. 

Comments

L71 – 72, EID50 = 10-2.5/0.1 mL :

How did the authors propagate and store CAV and GyH1? Describe it concretely. 

I cannot understand the meaning of “EID50 = 10-2.5/0.1 mL”.  Did they mean the titer of stock viruses? If so, the expression of “EID50 = 10-2.5/0.1 mL” is not correct.

Describe how the authors determined the titer (EID50) of CAV and GyH1 and also what method was used to determine the titre. 

Spell out “EID50”. 

2.2. Experimental infection and … , and Fig. A :

 The authors mentioned that the virus was inoculated into SPF chicken embryos by allantoic cavity inoculation (L80 – 81). However, to my understanding, the allantoic cavity inoculation has not been generally used to propagate and assay CAV, because CAV does not propagate in allantoic cavities. Usually, yolk sac inoculation is used to propagate and assay CAV, but moderate virus yields are obtained (See a review, Diseases of Poultry). According to Fig.1A (schematic diagram of experimental infection and testing), the authors observed chickens after hatch of embryos inoculated with the virus by allantoic cavity route. One of my questions is why the authors did not use one-day-old chicks which are known to be most susceptible to CAV. Explain the reasons why embryonated eggs and allantoic cavity inoculation were used instead of chicks.

Similarly, I wonder if allantoic cavity inoculation is the more sensitive method to propagate and assay GyH1 compared with chicken inoculation. Has the allantoic cavity inoculation route been used so far? If so, quote references if any. 

   In Fig. A, the authors mentioned that the viruses were inoculated into allantoic cavity of 6 Ed (day 6 embryo?). However, the embryo age was not described (L80). Spell out “Ed”. 

Moreover, it is stated that “CAV or GyH1 mono-infected chickens (not embryo) were inoculated with... CAV or GyH1... “ and also ”Co-infected chickens were inoculated with..... CAV and GyH1...” (L82 – 84). Were infected or co-infected chickens inoculated with the viruses again?  

In L127, the authors stated “Six-day-old SPF chickens were infected with CAV and/or GyH1”. Did the authors mean that chickens hatched from embryonated eggs infected by allantoic cavity inoculation were infected with the viruses at six days old again? I am totally confused. The authors should correctly describe the experimental procedures.

2.3. ELISA

Did the authors develop GyH1-ELISA in their laboratory? If so, describe how they created the ELISA system.

“serum virus” is a proper technical term? I have not heard so far (L161–162)

In this study, VP1 antigens in sera were detected by ELISAs. The authors should describe the data more concretely. For example, “the positive rate of VP1 antigens in serum...”, etc.

L162–164:

The description of this part is ambiguous and does not accord with Figs. 2A and 2B.

In L162–163, the authors mentioned that “the positive rate ....in co-infected chicken viremia decreased ...at 28 and 21 dpi,..”. However, the positive rate started to decrease from 21 dpi until 35 dpi (Figs. 2A and 2B).  

In L163–164, the authors mentioned that “while CAV or GyH1 serum positive rates did not decrease....... in CAV and GyH1 co-infected chickens”. I am confused about this description. In Line 162–163 , they stated “the positive rate .... in co-infected chicken viremia decreased”. Thus, different results about the positive rates are mentioned “in co-infected chickens”.

The results in mono-infected chicken groups are not given in the text. However, according to Figs. 2A and 2B, it seems that the CAV or GyH1 positive rate decreased in each mono-infected group more rapidly than in co-infected chickens after 21 dpi. Is this interpretation correct?

L167–168:

The authors should clearly define two infected groups (mono-, and co-infected) in description. Use “CAV or GyH1 mono-infected chickens” instead of “CAV or GyH1-infected chickens” throughout MS.  

L168–170:

I cannot understand the meaning of the sentence “The viral shedding in GyH1-infected chickens was not significantly increased in co-infected chickens before 21 dpi,..”. What do the authors mean by this sentence?  The viral shedding in GyH1 (mono)-infected chickens... .. in co-infected chickens?

According to Fig. C, viral (GyH1) shedding was probably significantly increased at 14 dpi (before 21 dpi) compared with 7 dpi in co-infected chickens as well as mono- infected chickens. Also, the viral shedding decreased after 21 dpi (Fig. C), although the authors stated “increased” (L169-170). Thus, they did not describe the results correctly.

If viremia and virus shedding in co-infected group are significantly more severe than in mono-infected group, indicate which data were statistically significant in Fig. 2 with * and P values like Fig.4.

Fig. 2B, legend: 

“plasma”→”serum”

3.3. Erythrocyte examination in the blood

Was the number of basophilic erythrocyte in mono-infected chickens significantly increased compared with mock chickens? 

How about between mono-infected and co-infected chickens? Was the increased number of basophilic erythrocyte in co-infected chickens statistically significant compared with mono-infected chickens?

Fig. 3 legend:

Fig. 3A should be explained in more detail. What do the red circles indicate?

3.4. Dynamic changes ......

According to Fig. 4, Fig. 4B is “Thymus”, and 4C is “Bursa”, which is not identical to the description in L215–216. 

Put “organ” between “immune” and “indexes” (L219–220). 

The authors repeated very similar statements in L215–216 and L220–221. Combine the sentences and state more clearly. 

(L221–222) The authors stated “caused severe immunosuppression”. Give specific evidence (data) on severe immunosuppression. Atrophy of immune organs and bone marrow may cause immunosuppression (possibility). However, the authors cannot state it definitively without indicating specific results of immunosuppression examined in their study. The statement is not a result and should be moved to the Discussion section. 

3.5. Viral loads in various tissues.......

Figs. 5A–C indicated the results of GyH1 viral loads but not CAV (L238–240). The authors should state the results in the text according to the order of Figs. 5A–C (GyH1) and 5D–F (CAV). 

(L242–244) Bursa in addition to gizzard, bone marrow, and thymus in co-infected chickens had also significantly higher GyH1 viral loads than in GyH1 mono-infected chickens at 21 dpi (Fig. 5B). Correct the statement. Similarly, the description in L244–245 is not correct. As mentioned above, at 21 dpi “bursa“ is also included in promoting GyH1 viral loads. Why did not the authors mention the results at 35 dpi in L245–246? 

GyH1 significantly enhanced the CAV replication in all tissues at all stages of infection examined. However, CAV showed poor enhancement of GyH1 replication. Is it possible to discuss the reasons of this difference in the Discussion section. 

3.6. Histopathological results

In the text, Fig. 6 is not indicated. Insert “Fig. 6” in appropriate positions to explain the results. 

The authors did not mention “adipose tissues in bone marrow in co-infected chickens” (L272–273). Describe the lesions observed in co-infected chickens as well as in mono-infected chickens. 

Fig. 6 legend

There is no explanation of black arrows indicated in the Gizzard picture. 

Were histopathological findings written in the legend equally observed in all infected groups?  There are no mentions of infected groups.  

L318:

Although the authors mentioned “serious contamination of live vaccine, live CAV vaccines are considered safe (no contamination) in at least several countries where strict vaccine quality controls are conducted, to my understanding (See a review in “Diseases of Poultry”). Therefore, the authors should not claim that the contaminated vaccine causes the problems in prevention and control of CAV infection without describing countries including China where vaccines contaminated with CAV are problematic. 

L322–324:

The citation of references (Davidson et al., 2004; Rosenberger and Cloud, 1998) is wrong, because GyH1 was first reported in 2012 (L45–46).

L 325–326:

Specific data (evidence) on “immunosuppression” were not given in this study, although damage to immune organs is thought to cause “immunosuppression”. Therefore, the authors cannot claim that “co-infected chickens exhibited more severe immunosuppression”.  Reconsider the discussion. 

L335:

Delete the citation of “McNeilly et al.”.

L379–380:

As mentioned above, the authors did not examine whether co-infection induced a more severe impact on “immunosuppressive ability” or not. Since co-infection caused more severe damage to immune organs and bone marrows, it is very likely that such damage induces more severe immunosuppression. However, in this study, the authors fail in giving specific immunological evidence on immunosuppression such as impaired humoral and/or cellular immune responses.

Author Response

Response to reviewers

We gratefully thank the editor and all reviewers for their time spend making their constructive remarks and useful suggestions, which has significantly raised the quality of the manuscript and has enable us to improve the manuscript. Each suggested revision and comment, brought forward by the reviewers was accurately incorporated and considered. Some inaccurate words have been revised in the manuscript. Below the comments of the reviewers are response point by point and made changes to the corresponding position in the manuscript. We sincerely hope that you find our responses and modifications satisfactory.

Point 1:Comments

L71 – 72, EID50 = 10-2.5/0.1 mL :

How did the authors propagate and store CAV and GyH1? Describe it concretely. 

I cannot understand the meaning of “EID50 = 10-2.5/0.1 mL”.  Did they mean the titer of stock viruses? If so, the expression of “EID50 = 10-2.5/0.1 mL” is not correct.

Describe how the authors determined the titer (EID50) of CAV and GyH1 and also what method was used to determine the titre. 

Spell out “EID50”. 

Response 1:GyH1 strain SDAU-1 (log10 EID50 = 2.5; GenBank accession no. MG366592) and CAV strain Cux-1 (log10 EID50 = 3; GenBank accession no. MN079036) was serially passaged in specific pathogen-free (SPF) chickens. We have also supplemented it in the manuscript.

As you said, the expression of “ EID50 = 10-2.5 / 0.1mL”  is not clear, and the expression of “log10 EID50 = 2.5” is more intuitive. Sincerely thank you for your rigorous consideration.

50% egg infectious dose (EID50) has been widely used in virology as an indicator of virus infective ability and virus titer. By inoculating a series of equally-diluted virus diluents into chicken embryos, the Reed-Muench method was used to calculate the number of eggs that were positive after the multiplication of the virus, and the dilution ratio that made 50 percent of chicken embryos positive could be obtained.

 Point 2:2.2. Experimental infection and … , and Fig. A :

 The authors mentioned that the virus was inoculated into SPF chicken embryos by allantoic cavity inoculation (L80 – 81). However, to my understanding, the allantoic cavity inoculation has not been generally used to propagate and assay CAV, because CAV does not propagate in allantoic cavities. Usually, yolk sac inoculation is used to propagate and assay CAV, but moderate virus yields are obtained (See a review, Diseases of Poultry). According to Fig.1A (schematic diagram of experimental infection and testing), the authors observed chickens after hatch of embryos inoculated with the virus by allantoic cavity route. One of my questions is why the authors did not use one-day-old chicks which are known to be most susceptible to CAV. Explain the reasons why embryonated eggs and allantoic cavity inoculation were used instead of chicks.

Similarly, I wonder if allantoic cavity inoculation is the more sensitive method to propagate and assay GyH1 compared with chicken inoculation. Has the allantoic cavity inoculation route been used so far? If so, quote references if any. 

   In Fig. A, the authors mentioned that the viruses were inoculated into allantoic cavity of 6 Ed (day 6 embryo?). However, the embryo age was not described (L80). Spell out “Ed”. 

Response 2:Based on your careful comments we found our terminology error. As shown in Figure 1A, needles were inserted in the median direction of the air chamber, not the boundary of the air chamber (allantoic cavity). The method we adopted was yolk sac inoculation, which has been modified in the manuscript.

The purpose of embryo egg inoculation instead of chicken inoculation is to provide practical reference value for our follow-up research and clinical practice.

We are sorry that we did not test the comparative experiment on embryo egg inoculation and chick inoculation with GyH1. At the early stage of the experiment (that is, when determining EID50), we also used embryonic egg inoculation and successfully measured 50% egg infectious dose.

“Ed” has been spelled out according to your opinion.

Point 3:Moreover, it is stated that “CAV or GyH1 mono-infected chickens (not embryo) were inoculated with... CAV or GyH1... “ and also ”Co-infected chickens were inoculated with..... CAV and GyH1...” (L82 – 84). Were infected or co-infected chickens inoculated with the viruses again?  

In L127, the authors stated “Six-day-old SPF chickens were infected with CAV and/or GyH1”. Did the authors mean that chickens hatched from embryonated eggs infected by allantoic cavity inoculation were infected with the viruses at six days old again? I am totally confused. The authors should correctly describe the experimental procedures.

Response 3: We feel sorry for the inconvenience brought to the reviewer. The infected or co-infected chickens were not inoculated with the virus again at other times, only at 6 days of embryonic development. We've changed the word "chickens" to "embryo".

Point 4:2.3. ELISA

Did the authors develop GyH1-ELISA in their laboratory? If so, describe how they created the ELISA system.

“serum virus” is a proper technical term? I have not heard so far (L161–162)

Response 4: References to our laboratory's creation of the GyH1-ELISA system have been inserted into the manuscript.

The optimal reaction conditions obtained by GyH1 DAS-ELISA are summarized as follows. ELISA plates were diluted to a final concentration of 3 µg/mL with 100 μL purified 2F2 monoclonal antibody in a coated buffer (CBS, pH 9.4) and incubated overnight at 4°C. The plates were sealed with 200µL 4% skim milk at 37℃ for 1 h, then washed 5 times with PBST. Subsequently, 100 µL GyH1-positive serum samples diluted with PBS to 1:4 were added to each well (in triplicated) and incubated at 37℃ for 1 h. Positive and negative controls were loaded into each plate. After washing with PBST for 5 times, the plates were incubated at 37℃ with 100 µL 6 µg/mL of detection antibody for 1 h. After 5 washes with PBST, 100 µL TMB (3,3', 5,5' -tetramethylbenzidine) was added to the plate for color development. After incubation at 37℃ for 15 min, 100 µL 2 mol/L H2SO4 was added to stop color development. After 5 min at room temperature, OD450 values were measured using an ELISA readout.

"serum virus" has been modified to "the positive rate of virus in serum".

Point 5:L162–164:

In this study, VP1 antigens in sera were detected by ELISAs. The authors should describe the data more concretely. For example, “the positive rate of VP1 antigens in serum...”, etc.

The description of this part is ambiguous and does not accord with Figs. 2A and 2B.

In L162–163, the authors mentioned that “the positive rate ....in co-infected chicken viremia decreased ...at 28 and 21 dpi,..”. However, the positive rate started to decrease from 21 dpi until 35 dpi (Figs. 2A and 2B).  

In L163–164, the authors mentioned that “while CAV or GyH1 serum positive rates did not decrease....... in CAV and GyH1 co-infected chickens”. I am confused about this description. In Line 162–163 , they stated “the positive rate .... in co-infected chicken viremia decreased”. Thus, different results about the positive rates are mentioned “in co-infected chickens”.

Response 5:The VP1 antigen has been described in detail according to your suggestion.

We are very sorry about this and thank you for your careful consideration. We have reversed the “mono-infection” and “co-infection”, which has been revised in the manuscript.

Point 6:The results in mono-infected chicken groups are not given in the text. However, according to Figs. 2A and 2B, it seems that the CAV or GyH1 positive rate decreased in each mono-infected group more rapidly than in co-infected chickens after 21 dpi. Is this interpretation correct?

Response 6:The above questions are reasonable. “it seems that the CAV or GyH1 positive rate decreased in each mono-infected group more rapidly than in co-infected chickens after 21 dpi” because CAV or GyH1 is present in serum for a short time in mono-infected chickens. So it's reasonable. The slower decline in the positive rate of CAV or GyH1 in co-infected chickens was due to the persistence of the virus in the chickens, that is, the viraemia was more severe in the flock after 28dpi.

Point 7:L167–168:

The authors should clearly define two infected groups (mono-, and co-infected) in description. Use “CAV or GyH1 mono-infected chickens” instead of “CAV or GyH1-infected chickens” throughout MS.  

Response 7:Thank you very much for your careful advice !

Point 8:L168–170:

I cannot understand the meaning of the sentence “The viral shedding in GyH1-infected chickens was not significantly increased in co-infected chickens before 21 dpi,..”. What do the authors mean by this sentence?  The viral shedding in GyH1 (mono)-infected chickens... .. in co-infected chickens?

According to Fig. C, viral (GyH1) shedding was probably significantly increased at 14 dpi (before 21 dpi) compared with 7 dpi in co-infected chickens as well as mono- infected chickens. Also, the viral shedding decreased after 21 dpi (Fig. C), although the authors stated “increased” (L169-170). Thus, they did not describe the results correctly.

If viremia and virus shedding in co-infected group are significantly more severe than in mono-infected group, indicate which data were statistically significant in Fig. 2 with * and P values like Fig.4.

Response 8:We gratefully appreciate for your valuable suggestion and these have been modified in MS.

Point 9:Fig. 2B, legend: 

“plasma”→”serum”

Response 9:Thank you very much.

Point 10:3.3. Erythrocyte examination in the blood

Was the number of basophilic erythrocyte in mono-infected chickens significantly increased compared with mock chickens? 

How about between mono-infected and co-infected chickens? Was the increased number of basophilic erythrocyte in co-infected chickens statistically significant compared with mono-infected chickens?

Response 10:Proerythrocyte should not be present in normal peripheral blood, but only appear when the hematopoietic system is diseased. The proerythrocyte was found in peripheral blood of mono-infected chickens, so they had more than the mock chickens.

Proerythrocyte accounted for 1.2% and 0.9% of the complete blood cell in CAV and GyH1 mono-infected chickens, respectively, while those in co-infected chickens accounted for 7.5%. These are significant.

Point 11:Fig. 3 legend:

Fig. 3A should be explained in more detail. What do the red circles indicate?

Response 11:It has been added in MS.

Point 12:3.4. Dynamic changes ......

According to Fig. 4, Fig. 4B is “Thymus”, and 4C is “Bursa”, which is not identical to the description in L215–216. 

Put “organ” between “immune” and “indexes” (L219–220). 

The authors repeated very similar statements in L215–216 and L220–221. Combine the sentences and state more clearly. 

(L221–222) The authors stated “caused severe immunosuppression”. Give specific evidence (data) on severe immunosuppression. Atrophy of immune organs and bone marrow may cause immunosuppression (possibility). However, the authors cannot state it definitively without indicating specific results of immunosuppression examined in their study. The statement is not a result and should be moved to the Discussion section. 

Response 12:Thanks for your careful advice, we have changed "immunosuppression" to "atrophy of immune organs", which is more objective.

Point 13:3.5. Viral loads in various tissues.......

Figs. 5A–C indicated the results of GyH1 viral loads but not CAV (L238–240). The authors should state the results in the text according to the order of Figs. 5A–C (GyH1) and 5D–F (CAV). 

(L242–244) Bursa in addition to gizzard, bone marrow, and thymus in co-infected chickens had also significantly higher GyH1 viral loads than in GyH1 mono-infected chickens at 21 dpi (Fig. 5B). Correct the statement. Similarly, the description in L244–245 is not correct. As mentioned above, at 21 dpi “bursa“ is also included in promoting GyH1 viral loads. Why did not the authors mention the results at 35 dpi in L245–246? 

GyH1 significantly enhanced the CAV replication in all tissues at all stages of infection examined. However, CAV showed poor enhancement of GyH1 replication. Is it possible to discuss the reasons of this difference in the Discussion section. 

Response 13:The above content has been modified.(L244-254).

"CAV showed poor enhancement of GyH1 replication." This is also a point that we originally wanted to discuss and we have added to the discussion.

Point 14:3.6. Histopathological results

In the text, Fig. 6 is not indicated. Insert “Fig. 6” in appropriate positions to explain the results. 

The authors did not mention “adipose tissues in bone marrow in co-infected chickens” (L272–273). Describe the lesions observed in co-infected chickens as well as in mono-infected chickens. 

Response 14:The above problems have been modified.

Point 15:Fig. 6 legend

There is no explanation of black arrows indicated in the Gizzard picture. 

Were histopathological findings written in the legend equally observed in all infected groups?  There are no mentions of infected groups.  

Response 15:The pathological changes shown in the figure are widely seen in each infected group.

Point 16:L318:

Although the authors mentioned “serious contamination of live vaccine, live CAV vaccines are considered safe (no contamination) in at least several countries where strict vaccine quality controls are conducted, to my understanding (See a review in “Diseases of Poultry”). Therefore, the authors should not claim that the contaminated vaccine causes the problems in prevention and control of CAV infection without describing countries including China where vaccines contaminated with CAV are problematic. 

Response 16:We should not discuss "CAV live vaccine contamination problem" without evidence, so we have deleted it.

Point 17:

L322–324:The citation of references (Davidson et al., 2004; Rosenberger and Cloud, 1998) is wrong, because GyH1 was first reported in 2012 (L45–46).

L335:Delete the citation of “McNeilly et al.”.

Response 17:These erroneous documents have been deleted.

Point 18:L 325–326:

Specific data (evidence) on “immunosuppression” were not given in this study, although damage to immune organs is thought to cause “immunosuppression”. Therefore, the authors cannot claim that “co-infected chickens exhibited more severe immunosuppression”.  Reconsider the discussion. 

L379–380:

As mentioned above, the authors did not examine whether co-infection induced a more severe impact on “immunosuppressive ability” or not. Since co-infection caused more severe damage to immune organs and bone marrows, it is very likely that such damage induces more severe immunosuppression. However, in this study, the authors fail in giving specific immunological evidence on immunosuppression such as impaired humoral and/or cellular immune responses.

Response 18:We modify "immunosuppression" to "atrophy of immune organs". And the discussion suggests that it may, but not necessarily will, cause more severe immunosuppression.

Thank you again for your valuable advice ! We hope our modification can satisfy the reviewers.

Reviewer 2 Report

Mengzan Yan et al. submitted the paper titled “The synergy of chicken anemia virus and gyrovirus homsa 1 in chickens” that described CAV and GyH1 synergistically inhibited weight gain, increased mortality and hindered erythropoiesis, and synergistically caused immune organs and digestive organs damages, and led to higher viral loads and shedding levels in co-infected chickens. So, they concluded that CAV and GyH1 synergistically promote immunosuppression, pathogenicity, and viral replication in co-infected chickens. CAV and GyH1 are both belong to Gyrovirus genus of Anelloviridae. The paper reported the synergy between CAV and GyH1 for the first time, which is of great significance to the study of the synergistic effect of Gyroviruses. However, there are some minor issues need to be revised as follow:

In PDF version

1.  “Anelloviridae” and “Circoviridae” in the paper should be written in italics.

2.  Line 82-84, Are you sure that the coinfection group chickens were inoculated with 100 EID50 of CAV and GyH1 in 0.1 mL PBS? While the single infection group chickens were inoculated with 100 EID50 of the CAV or GyH1 in 0.2 mL PBS.

3.  Line 90, What is “routine blood examinations”? Please explain it.

4.  Line 102, is the DAS-ELISA a kit? Or it come from where?

5.  Please regulate the Table 1, too long. And, the CAV in the table should not be boldface

6.  Please improve the quality of Fig. 1A, and remark the B, C and D.

7.  Line 163, “28 and 21 dpi” should be “21 and 28 dpi”.

8.  Please regulate the location of Fig 2, Fig4, Fig5 and Supplementary Fig 1, and remark the A, B, C and D et al.

9.  The Fig 3 cross the two pages, so it cannot be seen fully.

10.Please regulate the format according to “Viruses” require.

Author Response

Response to reviewers

We gratefully thank the editor and all reviewers for their time spend making their constructive remarks and useful suggestions, which has significantly raised the quality of the manuscript and has enable us to improve the manuscript. Each suggested revision and comment, brought forward by the reviewers was accurately incorporated and considered. Below the comments of the reviewers are response point by point and made changes to the corresponding position in the manuscript. We sincerely hope that you find our responses and modifications satisfactory.

Point 1:  “Anelloviridae” and “Circoviridae” in the paper should be written in italics.

Response 1:We gratefully appreciate for your valuable comment. The font style has been changed.

Point 2:  Line 82-84, Are you sure that the coinfection group chickens were inoculated with 100 EID50 of CAV and GyH1 in 0.1 mL PBS? While the single infection group chickens were inoculated with 100 EID50 of the CAV or GyH1 in 0.2 mL PBS.

   Response 2:Yes, we're sure it is. In order to determine the respective viral titers and infectivity of CAV and GyH1, EID50 was determined in our previous work. Co-infected chickens were inoculated with 100 EID50 of CAV and 100 EID50 GyH1 in 0.1 mL PBS.

Point 3:  Line 90, What is “routine blood examinations”? Please explain it.

Response 3:Blood routine examination is also called complete blood count(CBC).CBC is the test of the hemocyte count using a blood cell analyzer.

Point 4:  Line 102, is the DAS-ELISA a kit? Or it come from where?

Response 4:The DAS-ELISA method used to detect GyH1 was developed by our laboratory. I have added the corresponding references.

Point 5:  Please regulate the Table 1, too long. And, the CAV in the table should not be boldface

Response 5:Thank you for your suggestion. The form has been modified.

Point 6:  Please improve the quality of Fig. 1A, and remark the B, C and D.

Point 7:  Line 163, “28 and 21 dpi” should be “21 and 28 dpi”.

Point 8:  Please regulate the location of Fig 2, Fig4, Fig5 and Supplementary Fig 1, and remark the A, B, C and D et al.

Point 9:  The Fig 3 cross the two pages, so it cannot be seen fully.

Response 6-9:We sincerely appreciate your advice. Fig1A's picture definition has been improved. The position and format of the pictures in the manuscript have been standardized according to your valuable comments. We hope you are satisfied with our changes.

Point 10:  Please regulate the format according to “Viruses” require.

Response 10:Thank you for your valuable comments. We have modified the format of the reference materials according to the normative format of the viruses.
